# Prognostic Factors and Long-Term Outcome Prediction in Patients with Hypopharyngeal Carcinoma Treated with (Chemo)radiotherapy: Development of a Prognostic Model

**DOI:** 10.3390/biomedicines13020417

**Published:** 2025-02-09

**Authors:** Miloslav Pala, Pavla Novakova, Adam Tesar, Lucie Vesela, Antonin Vrana, Jarmila Sukova, Zdenka Pechacova, Petra Holeckova, Tereza Drbohlavova, Tomas Podlesak, Petra Tesarova

**Affiliations:** 1First Faculty of Medicine, Charles University in Prague, 12108 Prague, Czech Republic; adam.tesar@vfn.cz (A.T.); petra.holeckova@bulovka.cz (P.H.); petra.tesarova@bulovka.cz (P.T.); 2Institute of Radiation Oncology, Bulovka University Hospital, 18081 Prague, Czech Republic; lucie.vesela@bulovka.cz (L.V.); antonin.vrana@bulovka.cz (A.V.); jarmila.sukova@bulovka.cz (J.S.); zdenka.pechacova@bulovka.cz (Z.P.); tereza.drbohlavova@bulovka.cz (T.D.); 3Radiophysics Department, Bulovka University Hospital, 18001 Prague, Czech Republic; pavla.novakova@bulovka.cz; 4Department of Otorhinolaryngology, Bulovka University Hospital, 18001 Prague, Czech Republic; tomas.podlesak@bulovka.cz

**Keywords:** hypopharyngeal carcinoma, curative radiotherapy, chemoradiotherapy, prognostic factors, predictive model

## Abstract

**Background/Objectives**: To evaluate the effectiveness of curative (chemo)radiotherapy in patients with hypopharyngeal carcinoma and to identify prognostic factors influencing treatment outcomes. **Methods**: We conducted a retrospective study of 173 consecutive patients, treated with definitive or postoperative (chemo)radiotherapy from 2002 to 2020 [median age 60 years; current/former smokers 95%; UICC stage III/IV 96%]. Radiation therapy was preceded by a radical resection of a primary tumor in 32% of patients. One hundred patients received chemotherapy. **Results**: The median total dose of radiotherapy achieved was 70 Gy. The five- and ten-year locoregional controls were 63%, and the five- and ten-year distant controls were 77% and 76%, respectively. The five- and ten-year overall survival rates were 24% and 9%, respectively. **Conclusions**: The results demonstrate the limited effectiveness of curative (chemo)radiotherapy in patients with hypopharyngeal carcinoma with long-term locoregional and distant control of half of the treated patients. The multivariate analysis indicated that initial surgery, chemotherapy, comorbidity score (as assessed by ACE-27), pretreatment tracheostomy, hemoglobin level and initial response to treatment were the strongest prognostic factors in predicting survival. Using these factors, corresponding predictive models were constructed.

## 1. Introduction

Hypopharyngeal carcinomas are relatively rare, representing <1% of all solid tumors [1]. Patients with hypopharyngeal carcinoma are predominantly male and commonly have a history of tobacco and heavy alcohol consumption [2,3]. Hypopharyngeal carcinoma is among the most aggressive head and neck cancers. Due to the location, tumors can grow significantly before giving rise to any clinical symptoms. Hypopharyngeal carcinoma is generally characterized by an advanced stage at presentation with extensive submucosal spread, a high risk of regional lymphatic involvement and strong tendency to develop distant metastases, resulting in poor treatment outcomes [4,5,6].

The optimal treatment for hypopharyngeal carcinoma is controversial. Achieving maximum local control through radical treatment while minimizing its consequences is a considerable challenge when dealing with this group of tumors. Therapeutic decision-making includes consideration of stage, site, performance status, comorbidities and nutritional status in addition to personal and institutional preferences. Historically, surgery followed by radiation has been the standard of care for patients presenting with advanced disease. However, the substantial deterioration of quality of life after surgical resection has enhanced the development of organ preservation strategies. Larynx preservation therapy, with radiotherapy and chemotherapy, has been shown to produce equivalent survival rates to laryngectomy followed by radiotherapy [7,8,9]. Regarding treatment options, definitive surgery followed by radiotherapy or laryngeal preservation using chemoradiotherapy is considered the standard of care; however, there is continuing debate on the outcomes due to conflicting data.

Several factors related to patients, tumors and therapy affect the treatment results. Prognostic factors reported in retrospective studies are as follows: Age [10,11,12], performance status [13,14], comorbidities [15,16,17], body mass index [17], subsite [15,18,19], T-staging [10,11,20,21], N-staging [10,11,20,22,23], stage [15,19,24,25], esophageal invasion [26,27], tumor volume [28,29,30,31], extranodal extension [26,32,33], lymphatic invasion [19,31], radicality of resection [18,26,32,34], chemotherapy [21,35,36], initial response to treatment [21], HPV status [37,38], albumin level [17], hemoglobin level [39,40] and neutrophil-lymphocyte ratio [41].

In this study, we aim to analyze long-term treatment outcomes in a consecutive group of patients treated with definitive or postoperative (chemo)radiotherapy at the Institute of Radiation Oncology and identify prognostic factors that affect treatment results.

## 2. Materials and Methods

Over the period of January 2002 to April 2020, 223 patients were treated for hypopharyngeal carcinomas at the Institute of Radiation Oncology. One hundred seventy-three consecutive patients with hypopharyngeal carcinoma who started definitive or postoperative radiotherapy with a curative intent were included in the study. In total, 50 patients were excluded (23 were in palliative treatment for bad general conditions, 14 were categorized with metastatic disease, 9 were featuring a locoregionally recurrent tumor, and 4 had a synchronous tumor in the head neck region). The median follow-up of surviving patients was 60 months. The median age at the time of treatment initiation was 59 years (range 42–82). The female to male ratio was 1: 7.7. The vast majority of patients were smokers or former smokers (95%); approximately two-thirds of patients admitted to daily alcohol consumption. All tumors were retrospectively reclassified according to the seventh edition of UICC tumor node metastasis (TNM) Classification. The majority of patients were treated for locally advanced disease (96% UICC stage ≥ III). Regional cervical metastases were initially diagnosed in 85% of patients. Five patients with metastases to the upper mediastinal nodes (UICC stage IVC) treated with curative intent were included in the study. Squamous cell carcinoma was the most frequent histology (Table 1).

### Treatment

Surgery: In 56 patients (32%), radiotherapy was preceded by resection of the primary tumor; 54 of these patients underwent bilateral or unilateral neck dissection. A total of five patients underwent endoscopic resection for the primary tumor; other patients underwent open surgical approaches. Full radicality (resection margins ≥ 5 mm) was declared in 71% of patients who underwent resection. In eight patients, the surgical procedure was limited only to neck dissection without resection of the primary tumor. In the remaining patients, surgery was limited to biopsy verification (Table 2). Chemotherapy has been recommended in patients treated without initial surgery or postoperatively, depending on risk factors.

Radiotherapy: Patients with inoperable diseases or patients with severe comorbidities and patients refusing surgery were treated with definitive radiotherapy. Before 2007, patients were treated with 2D and 3D conformal radiotherapy (46 cases). Patients were treated with intensity-modulated radiation therapy (IMRT) from 2007 onwards (127 cases). The prescribed dose was 70 Gy/35 fractions for definitive radiotherapy or 64–70 Gy/32–35 fractions for postoperative radiotherapy (Table 2).

Chemotherapy: Chemotherapy has been recommended in patients treated without initial surgery or postoperatively, depending on risk factors. A total of 100 patients received systemic therapy, 97 of them concomitantly (cisplatin 40 mg/m^2^ qw in 92 patients, cisplatin 100 mg/m^2^ q3w in 2 patients, docetaxel 40 mg/m^2^ qw in 1 patient, cetuximab 400 → 250 mg/m^2^ qw in 2 patients). Six patients received neoadjuvant chemotherapy based on platinum derivatives (three of which received concomitant chemotherapy as well (Table 2).

Analysis: For statistical analysis, all data were recorded and analyzed on XLSTAT software (Addinsoft) version 18.07. Kaplan–Meier methods were used to estimate locoregional control (LRC), distant metastasis-free interval (DMFI), disease-free interval (DFI), overall survival (OS), and disease-free survival (DFS). The survival or disease-free periods counted from the start of radiation to the time of relapse (LRC, DMFI, DFI) or death (OS) or relapse and death (DFS). The log-rank test was used to compare survival and recurrence rates between various parameters. We used the Cox regression hazard model to analyze multivariate data. All analyses were performed with a two-sided significance level of ≤0.05. Acute and late toxicity were evaluated according to RTOG (Radiation Therapy Oncology Group) criteria [42]. Comorbidities present at the time of diagnosis were collected retrospectively using the ACE-27 index [43]. We used multinomial logistic regression to establish a predictive model. The overall survival/disease-free interval was discretized by year and was used as the ordinal response. Type discretization had no influence on the significance of the model. We tested chemotherapy, radiotherapy, initial surgery, ACE-27, alcohol intake, smoking, hemoglobin level, grading, staging, histology—radiotherapy interval, PEG insertion, response to treatment, pretreatment tracheostomy, gender and age as predictors. Finally, we created a scale scoring the most significant parameters. The overall score was then verified to be a significant predictor by univariate logistic regression. All the regression statistics were processed using MATLAB R2018b statistic tools (Math-Works) with a level of significance of (*p* = 0.05).

## 3. Results

### 3.1. Locoregional Control

A total of 65 locoregional failures were detected. In 37 patients, there was persistence after the end of treatment; 28 patients failed during the follow-up within 1–61 months after the end of radiotherapy. The vast majority of locoregional failures (98%) were detected in the first 36 months. The five- and ten-year locoregional controls were 63% (Figure 1).

A total of six patients underwent salvage surgery; five patients died 5–25 months later, while one patient survived 15 months after salvage surgery. Salvage reirradiation was performed in four patients; all died 5–13 months after detection of the recurrence. Fifteen patients (23%) were treated with palliative chemotherapy and died 4–48 months after recurrence. The remaining 40 patients received only symptomatic treatment.

### 3.2. Distant Control

Distant failure was reported in 41 patients 1–73 months after completion of radiotherapy; most occurred within 36 months after treatment—in 36 patients (88%). Distant metastases were most often detected in the lung area. The five- and ten-year distant controls were 77% and 75%, respectively (Figure 2).

Out of 41 patients with distant failure, five patients with metastases to the lungs were treated locally: two patients underwent surgical metastasectomy (subsequently dying 27 and 13 months after the failure), three patients underwent stereotactic radiotherapy (two of whom died 11 and 17 months after the failure, and one living 36 months after detection of metastases). One patient with liver metastases underwent radiofrequency ablation and lived 21 months. Fifteen patients underwent palliative chemotherapy, all of whom died within 1–21 months of evidence of metastases. Three patients underwent palliative radiotherapy (they died four, five and nine months after detection of metastases). The remaining seventeen patients received only symptomatic treatment.

### 3.3. Survival

A total of 146 patients died. The five- and ten-year overall survival rates were 23% and 9%, respectively (Figure 3). The five- and ten-year disease-free survival rates were 22% and 9%, respectively (Figure 4). Tumor progression was the primary cause of death in 87 patients. In 49 patients, the cause of death was unrelated to cancer. During the follow-up, four metachronous tumors in the head and neck area were diagnosed (all squamous cell carcinomas in the oral cavity 4–36 months after treatment); eleven metachronous duplicate tumors outside the head and neck area were diagnosed in ten patients 4–103 months after treatment. Duplicate tumor progression was the cause of death in ten of these cases.

### 3.4. Univariate and Multivariate Analysis

The parameters that reached statistical significance in the univariate analysis were as follows: initial surgery, comorbidity score, total time of radiotherapy, concomitant chemotherapy, pretreatment tracheostomy; concomitant amifostin, hemoglobin level and, initial response to treatment (Table 3).

The multivariate analysis of variables showed the following independent prognostic parameters: Initial surgery for OS (HR 0.659; 95% CI 0.439–0.991; *p* = 0.045), DFI (HR 0.556; 95% CI 0.331–0.934; *p* = 0.027) and DFS (HR 0.649; 95% CI 0.430–0.883; *p* = 0.039), moderate or severe comorbidities for DFS (HR 0.568; 95% CI 0.376–0.857; *p* = 0.007), pretreatment tracheostomy for OS (HR 0.620; 95% CI 0.413–0.930; *p* = 0.021), chemotherapy for DFS (HR 0.620; 95% CI 0.435–0.883; *p* = 0.008), hemoglobin level for OS (HR 0.637; 95% CI 0.449–0.906; *p* = 0.012), and initial response to treatment for OS (HR 0.129; 95% CI 0.080–0.208; *p* < 0.0001), DFI (HR 0.008; 95% CI 0.002–0.025; *p* < 0.0001) and DFS (HR 0.027; 95% CI 0.013–0.055; *p* < 0.0001) (Table 4).

### 3.5. Predictive Model

The multinomial logistic regression analysis in accordance with Cox proportional hazards regressions identified the predictors as OS initial operation and initial treatment response, but also chemotherapy. For DFI, we identified as predictors the initial operation, chemotherapy and interval between histology and radiotherapy in days (Table 5). Then we created a scoring scale (Hypopharyngeal risk score) to estimate the probability of OS/DFI in clinical practice. The score is a linear combination of these parameters (operation: YES—0 point, NO—1 point; chemotherapy: YES—0 point, NO—1 point; tracheostomy: NO—0 point, YES—1 point; hemoglobin level below 100 g/L: NO—0 point, YES—1 point; initial treatment response: YES—0 point, NO—3 points). This score was verified as a significant predictor by univariate logistic regression analysis (Table 6) The predictive importance of our scale could be demonstrated by the dependence of the probability of OS/DFI in less than 2 years (median OS = 1.66 years, median DFI 1.88 years). Patients with a Hypopharynx risk score of 4 or more had a 0.88 probability to die before year two (Figure 5).

## 4. Discussion

The optimal treatment of hypopharyngeal carcinoma remains unknown. Due to the rarity of the disease, we have only limited data from prospective clinical studies, and we are thus dependent on retrospective work burdened by a low number of cases, the heterogeneous population of patients and inconsistency in treatment procedures. Patients with early cancer of the hypopharynx are most often treated with larynx-sparing surgery or definitive radiotherapy. In more advanced tumors, a surgical resection alone cannot guarantee sufficient locoregional control. Patients with a locoregionally advanced form of hypopharyngeal cancer are treated with multimodality procedures—radical surgery followed by postoperative radiotherapy ± chemotherapy or definitive radiotherapy ± chemotherapy. Retrospective studies report a five-year locoregional control in the range of 18–72%, distant control 57–79%, and a five-year overall survival rate between 15–53% [6,20,21,22,23,25,44,45,46,47,48,49,50,51,52,53].

An analysis of SEER (the Surveillance, Epidemiology, and End Results program) database patients (n = 857) did not show statistically significant differences in the overall survival rate in patients treated primarily with a surgical approach or with primary radiotherapy, even in the subgroup of patients with T4a tumors [11]. A study by Japanese authors (n = 254) also showed no statistically significant differences between primary surgery and postoperative radiotherapy or definitive chemoradiotherapy in overall survival, and even here, no significant difference was noted in the subgroup of patients with T4a tumors [46]. The negative impact of larynx preservation protocols on oncological outcomes was not noted, even in other prospective and retrospective studies comparing primarily surgical and primarily non-surgical approaches [20,34,36,47,48,54,55,56,57,58,59,60]. Larynx preservation protocols in the form of induction chemotherapy followed by radiotherapy, or in the form of concomitant chemoradiotherapy, thus prove to be a suitable alternative in patients whose extent of tumor involvement requires a total laryngectomy.

Comparable effectiveness of both treatment approaches is not reported uniformly. A randomized trial by Beauvillain et al. compared the effectiveness of induction chemotherapy followed by radiotherapy (first arm) or by total laryngectomy and postoperative radiotherapy (second arm). This smaller study reported a worsening of local control (63% vs. 39%; *p* < 0.01) and five-year overall survival (37% vs. 19%; *p* = 0.04) in patients treated with a non-surgical larynx-preserving procedure [61]. A large-scale analysis of the SEER database, including nearly 4000 patients, also reports a worsening of overall survival rate in patients with hypopharyngeal cancer treated non-surgically compared to patients treated with a combination of surgery and radiotherapy [10]. Higher effectiveness of primarily radical surgery was also reported in some other retrospective evaluations [22,54,62].

Differences in the effectiveness of both primary procedures are particularly noticeable in locoregionally advanced carcinoma of UICC stage IV. Shirai et al. reported, in a retrospective evaluation (n = 101), better three-year survival results for patients with stage IVA treated primarily with surgery, while no statistically significant differences were noted between the two approaches for patients with stages I–III [24]. Significant differences in the effectiveness of both procedures in stage IV carcinoma were also noted in a retrospective evaluation by Yoon et al. (81% for two-year locoregional control in surgical procedure cases, 33% and 32% for radiotherapy and chemoradiotherapy cases, respectively; *p* = 0.006) [21]. The work of Tsai et al., including 652 clinical stage III/IV patients treated with primary surgery or concomitant chemoradiotherapy, reported a statistically significant worsening of overall survival in patients treated without initial surgical resection (33% vs. 45%; *p* < 0.001); the greatest benefit of initial surgery was demonstrated in clinical stage IVA [25]. An analysis of more than 9000 National Cancer Database (NCDB) patients also noted statistically significant differences in survival in favor of surgical treatment in T4 hypopharyngeal carcinoma [38]. Two recently published meta-analyses do not provide a clear conclusion. While a meta-analysis of seventeen clinical trials (n = 2539) comparing surgical and non-surgical procedures showed no difference in overall survival [63], a meta-analysis of thirteen clinical trials (n = 1994) reported a statistically significant improvement in local control and overall survival in patients treated with surgery followed by radiotherapy ± chemotherapy compared to definitive chemoradiotherapy [64].

In our study, initial resection of the primary tumor was performed in 56 patients (32%), of which 43 patients required a total laryngectomy. An initial surgical procedure was shown to be an independent positive prognostic factor for overall survival (HR 0.659; 95% CI 0.439–0.991; *p* = 0.045), disease-free interval (HR 0.556; 95% CI 0.331–0.934; *p* = 0.027) and disease-free survival (HR 0.649; 95% CI 0.430–0.883; *p* = 0.039) in multivariate analysis. The reason for the varying effectiveness is probably due to the high proportion of locoregionally advanced tumors in the group, where the absolute majority (82%) consisted of UICC stage IV. Our results are consistent with those of a recent meta-analysis by Tsai et al., in which the authors demonstrated that upfront surgery showed better survival than definitive concurrent chemoradiation [65].

Platinum-based chemoradiotherapy is now a standard component of a therapeutic algorithm in patients with squamous cell carcinoma of the head and neck [66]. The results of the landmark studies (RTOG 9501, EORTC 22931) led to a shift in postoperative treatment, and platinum-based chemoradiotherapy became the standard of care for postoperative high-risk locally advanced carcinomas [67,68]. Boehm et al. demonstrated a significant improvement in overall and tumor specific survival since adjustment of adjuvant postoperative treatment according to the landmark trials in advanced laryngeal and hypopharyngeal squamous cell carcinoma [69]. The appropriate sequence of chemotherapy was the subject of a randomized trial by Prades et al., which compared concomitant chemotherapy or induction chemotherapy followed by radiotherapy. Concomitant chemoradiotherapy demonstrated higher larynx preservation potential, without significant impact on the survival of treated patients [70]. The FNCLCC-GORTEC randomized trial tested an altered fractionated regimen of radiotherapy ± concomitant chemotherapy in unresectable cancers of the oropharynx and hypopharynx (n = 163). The results favored patients treated with concomitant chemoradiotherapy (two-year overall survival 38% vs. 20%), albeit at the cost of increased early toxicity with a higher need for nutritional gastrostomy [71].

In line with these data, we also note a significant prognostic impact of chemotherapy. Systemic treatment was given as part of the primary treatment in 58% of patients, mostly in the form of a concomitant weekly application of cisplatin 40 mg/m^2^. The evaluation demonstrated a statistically significant impact of the combination of radiotherapy and chemotherapy on disease-free survival (HR 0.620; 95% CI 0.435–0.883; *p* = 0.008). Univariate analysis indicated a trend in both improved disease-free interval (*p* = 0.099) and overall survival (*p* = 0.053). The results are consistent with the results of retrospective evaluations reporting the benefits of combined radiotherapy and chemotherapy for hypopharyngeal carcinomas [21,72].

The five-year overall survival rate of the entire cohort was only 23%. Non-tumor mortality contributed to it to a greater extent. A large proportion of patients were affected by severe comorbidities and elements of a self-destructive lifestyle. Various methodologies, including ACE-27, have repeatedly demonstrated the significant prognostic significance of comorbidities in patients with head and neck tumors [73,74]. In hypopharyngeal carcinoma, Tanaka et al. demonstrated the impact of comorbidities assessed using the Charlson comorbidity index on the overall and tumor-specific survival of patients treated with chemoradiotherapy [16]. Homma et al., in a retrospective evaluation of 156 patients, demonstrated a statistically significant worsening of survival in patients with moderate to severe comorbidities assessed by ACE-27 (45% vs. 28%) [15].

In our study, an ACE-27 score at the time of treatment entry proved to be a strong independent prognostic factor, and patients with scores of 0–1 achieved a statistically significant increase in disease-free survival compared to patients with scores of 2–3 (HR 0.568; 95% CI 0.376–0.857; *p* = 0.007); moreover, a trend toward worsening tumor control (*p* = 0.078) and overall survival (*p* = 0.082) in patients with an ACE score >1 was detected.

Solid tumors contain areas with a large number of hypoxic cells, with high heterogeneity both within a single tumor and between different tumors of the same histological type [75,76,77]. The processes leading to hypoxia are not fully elucidated, but probably involve various mechanisms of impaired oxygen delivery to the tumor microenvironment (vascular abnormalities, intratumoral pressure gradients, acute/chronic anemia) and its consumption [78]. Treatment results achieved by curative radiotherapy are negatively affected by reduced intratumoral oxygenation [79,80]. Experimental studies have shown that hypoxic cells are two to three times more resistant to one fraction of ionizing radiation compared to cells with normal oxygenation [79,80]. Reduced oxygenation increases tumor cell radioresistance [81,82] and increases both mutagenic potential and genetic instability [83,84]. Anemia (<120 g/L) is noted at the start of radiotherapy in up to 40–60% of irradiated patients. This proportion further increases to 80% during radiotherapy [80]. While the prognostic impact of the serum hemoglobin level on the results of radiotherapy in laryngeal cancer has been reported many times [85,86,87,88,89], such data are lacking in hypopharyngeal cancer. Fukada et al., in a multivariate analysis of 72 patients with locally advanced carcinoma of the oropharynx and hypopharynx treated with concomitant chemoradiotherapy, noted the impact of hemoglobin level on three-year disease-free survival [90]. The only work published so far that focused on hypopharyngeal carcinoma is the work by Suzuki et al. evaluating the results of non-surgical treatment of nineteen patients with T3-T4 hypopharyngeal cancer. In this small cohort, hemoglobin level (≥115 g/L) was an independent positive factor for 3-year locoregional control (90% vs. 34%), overall survival (70% vs. 33%), and progression-free survival (62% vs. 33%) [40]. In our study, a reduced hemoglobin level <100 g/L at the start and/or during radiotherapy proved to be an independent prognostic factor in multivariate analysis; patients with a hemoglobin level ≥100 g/L achieved statistically significantly longer survival than patients with a hemoglobin level <100 g/L (HR 0.637; 95% CI 0.449–0.906; *p* = 0.012).

A high percentage of patients with hypopharyngeal cancer have a tracheostomy inserted before starting radiotherapy. A permanent tracheostomy is an obvious part of the surgical procedure, including a total laryngectomy, while a prophylactic tracheostomy is performed when there is a risk of airway obstruction. Reports of the prognostic impact of a tracheostomy in patients with head and neck tumors are sparse and reported results are ambiguous. Mendenhall et al., in a cohort of 118 patients with T3 laryngeal carcinoma who were treated primarily with surgical and non-surgical procedures, did not note the effect of a pretreatment tracheostomy on locoregional control or tumor-specific survival [91]. Chiesa et al., in a small cohort of 21 patients with cancer of the larynx and hypopharynx treated non-surgically, also reported no differences in overall and tumor-specific survival between patients with and without a tracheostomy [92]. Nor was the tracheostomy shown to be a prognostically significant factor for laryngeal preservation, disease-free survival, and tumor-specific survival in patients with laryngeal cancer (n = 270) treated primarily non-surgically [93]. On the contrary, Herchenhorn et al. found that in a group of 49 patients with laryngeal tumors treated with definitive chemoradiotherapy, they demonstrated pretreatment tracheostomy as an independent negative prognostic factor for progression-free survival (HR 2.83, CI 95% 1.60–4.88, *p* < 0.001) and overall survival (HR 2.37, CI 95% 1.43–3.93, *p* < 0.001) [94]. Tennat et al. also reported a worsening of annual mortality in a group of 60 patients with stage III/IV cancer of the larynx and hypopharynx treated with a larynx-preservation procedures [95].

The prognostic impact of pretreatment tracheostomy has not yet been reported exclusively for the group with hypopharyngeal carcinomas. In our study, a tracheostomy was performed before starting radiotherapy in 71% of patients. Pretreatment tracheostomy was shown to be an independent prognostic factor in multivariate analysis; patients who started radiotherapy with a tracheostomy had statistically significantly worse overall survival than patients who started radiation without a tracheostomy (HR 0.620; 95% CI 0.413–0.930; *p* = 0.021). Univariate analysis indicated a trend in both worsening tumor control (*p* = 0.015) and disease-free survival (*p* = 0.009) in patients with pretreatment tracheostomy.

Prediction of overall survival and disease-free interval would be welcome for advance care planning in patients. It would enable better decision making for patients in case of a wish to withdraw treatment. In the future, this also could be crucial to stratify the exceptionally risky set of patients for different or even more aggressive treatments. Timing of palliative interventions or therapy withdrawal could be adjusted according to our prediction, and nutrition goals could be changed in case of very short survival or in patients with longer life expectancy. All patients were referred to our clinic for radiotherapy; therefore, our model can only be used for patients treated with radiotherapy. Another limitation of our predictive model is the use of unicentric data from a relatively small and very homogeneous group of patients. Therefore, it should be used only as a rough estimation; before clinical applicability, predictions of our model must be evaluated on a larger and more diverse population, as external validation is currently lacking. In such a case, even our scale (Hypopharyngeal risk score) or a more complex clinical scale could be suitable for future planning.

Limitations of our study include its retrospective nature and heterogeneity. Selection bias and treatment bias occur in retrospective, non-randomized patient cohorts, and these biases cannot be controlled for in statistical analyses. The patients were treated over nearly two decades, during which radiation techniques, surgical methods, and systemic therapies evolved. In the study, all consecutive patients with hypopharyngeal carcinoma who started definitive or postoperative radiotherapy with a curative intent were included. These changes might have contributed to the results we observed. Consequently, the results should be interpreted within the context of the limitations of a retrospective study.

## 5. Conclusions

The results of the retrospective study demonstrate the limited effectiveness of curative definitive and postoperative (chemo)radiotherapy in patients treated for hypopharyngeal carcinoma. In our cohort of consecutive patients, where the majority of treated tumors consisted of UICC stage IV, locoregional and distant control was achieved in half of the treated patients. The five- and ten-year locoregional controls were 63%, and the five- and ten-year overall survival rates were 24% and 9%, respectively. The multivariate analysis identified that the initial radical surgery, comorbidity score (as evaluated by ACE-27), pretreatment tracheostomy, hemoglobin level < 100 g/L and initial response to treatment are the strongest prognostic factors in predicting survival.

## Figures and Tables

**Figure 1 biomedicines-13-00417-f001:**
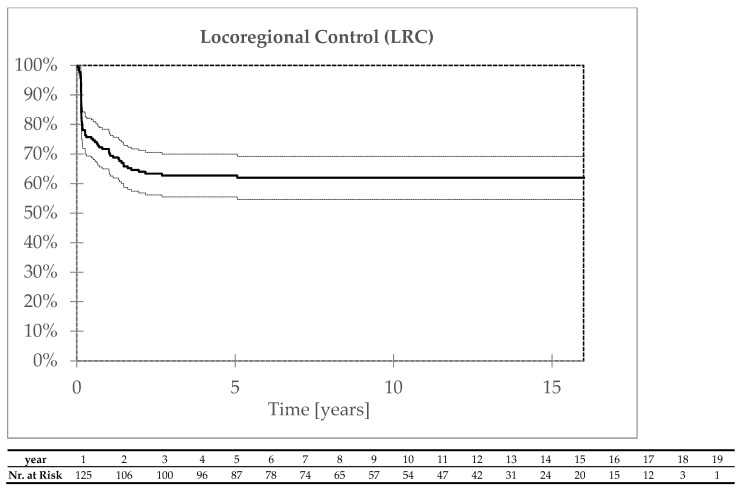
Locoregional control.

**Figure 2 biomedicines-13-00417-f002:**
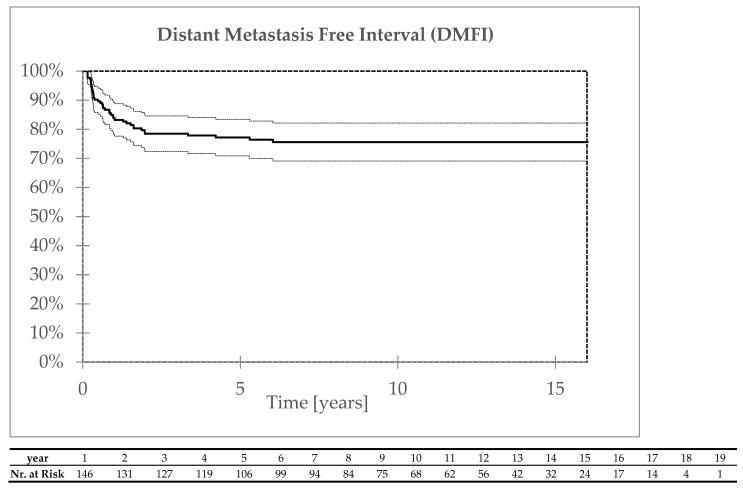
Distant control.

**Figure 3 biomedicines-13-00417-f003:**
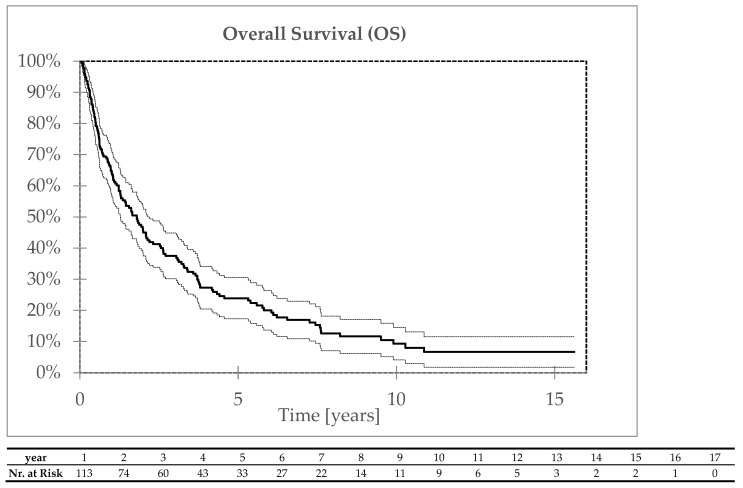
Overall survival.

**Figure 4 biomedicines-13-00417-f004:**
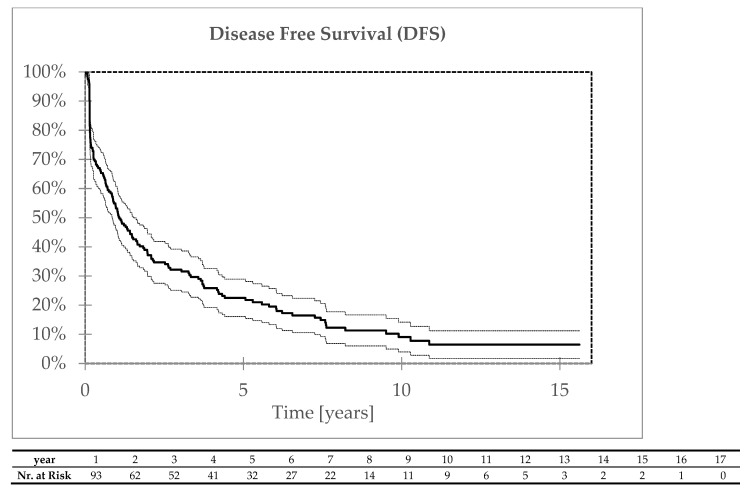
Disease-free survival.

**Figure 5 biomedicines-13-00417-f005:**
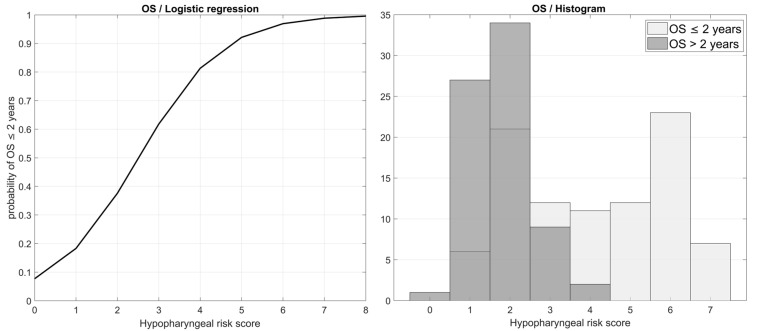
Logistic regression of overall survival of less than 2 years by Hypopharyngeal risk score as predictor.

**Table 1 biomedicines-13-00417-t001:** Demographic and tumor characteristics. Abbreviations: ACE = adult comorbidity evaluation score; G = grading.

Parameter	n	%
Age (years)	(42–82)	
Median	59	
Sex		
Males	153	88
Females	20	12
Smoking		
Chronic nicotinism	140	81
Former (>5 years)	25	14
Non-smoker	5	3
Unknown	3	2
Alcohol		
Daily	121	70
Occasionally	37	21
None	11	6
Unknown	4	2
Comorbidities		
ACE 0	59	34
ACE 1	81	47
ACE 2	25	14
ACE 3	8	5
Subsite		
Piriform sinus	157	91
Posterior pharyngeal wall	13	7
Post-cricoid region	3	2
T-staging		
T1	22	13
T2	36	21
T3	50	29
T4a	55	32
T4b	10	6
N-staging		
N0	26	15
N1	25	14
N2a	9	5
N2b	70	40
N2c	28	16
N3	15	9
M-staging		
M0	168	97
M1	5	3
Clinical stage		
I	3	2
II	3	2
III	26	15
IVA	117	68
IVB	19	11
IVC	5	3
Histological type		
Epidermoid carcinoma	165	95
Undifferentiated carcinoma	7	4
Sarcomatoid carcinoma	1	1
Grading		
G1	8	5
G2	86	50
G3/4	59	34
Unknown	20	12

**Table 2 biomedicines-13-00417-t002:** Treatment. Abbreviations: 2D/3D-CRT = two-dimensional/three-dimensional conformal radiotherapy; IMRT = intensity modulated radiotherapy.

Treatment	n	%
Surgery		
Radical surgery of primary	56	32
Neck dissection only	8	5
No surgery	109	63
Type of surgery	56	100
Endoscopically	5	9
Open resection	51	91
Radicality of resection	56	100
R0 (≥5 mm)	40	71
R0 (> 1 < 5 mm)	4	7
R1 (0 ≤ 1 mm)	10	18
R2	1	2
RX	1	2
Neck dissection	62	100
Unilateral	47	76
Bilateral	15	24
Radiotherapy		
Postoperative	56	32
Definitive	117	68
Radiotherapy technique		
2D/3D-CRT	46	27
IMRT	127	73
Total irradiation dose (Gy)	(18–72)	
Median	70	
Mean	67.3	
Chemotherapy		
Concomitant	94	54
Neoadjuvant + concomitant	3	2
Neoadjuvant	3	2
No chemotherapy	73	42

**Table 3 biomedicines-13-00417-t003:** Univariate cox proportional hazards regressions analyses for overall survival (OS), disease-free interval (DFI), and disease-free survival (DFS). Abbreviations: G = grading; ACE = adult comorbidity evaluation score; Hb = hemoglobin level; CR = complete response; HR = hazard ratio; CI = confidence interval.

Parameter	Groups	OS			DFI			DFS		
		HR	95%CI	*p*-Value	HR	95%CI	*p*-Value	HR	95%CI	*p*-Value
Age	≤65 vs. >65 years	0.780	0.543–1.120	0.177	1.111	0.710–1.738	0.645	0.882	0.618–1.258	0.488
Sex	female vs. male	1.317	0.768–2.258	0.316	1.356	0.656–2.805	0.409	1.378	0.804–2.361	0.242
Education	higher vs. basic	1.079	0.705–1.652	0.128	0.948	0.551–1.630	0.827	1.097	0.717–1.678	0.109
Marrital status	married vs. others	1.260	0.906–1.751	0.387	1.225	0.802–1.870	0.631	1.300	0.938–1.803	0.284
T category	T1–2 vs. T3–4	1.348	0.951–1.913	0.093	1.568	0.581–1.886	0.055	1.350	0.955–1.908	0.088
N category	N0 vs. N+	1.410	0.886–2.246	0.145	1.047	0.715–2.170	0.878	1.220	0.780–1.908	0.383
Stage (UICC)	I–III vs. IV	1.237	0.804–1.906	0.332	1.246	0.889–2.140	0.437	1.125	0.741–1.709	0.580
Grading	G1/2 vs. G3	1.264	0.881–1.812	0.391	1.379	0.950–2.491	0.331	1.335	0.936–1.904	0.277
Comorbidities	ACE 0–1 vs. 2–3	1.427	0.953–2.136	0.082	1.538	0.582–1.719	0.078	0.640	0.954–0.430	0.027
Smoking	non-smoker vs. smoker	0.887	0.582–1.349	0.282	1.001	0.568–1.418	0.337	0.926	0.609–1.408	0.256
Alcohol	no/occasionally vs. daily	1.099	0.765–1.580	0.332	0.898	0.961–2.242	0.209	1.098	0.765–1.576	0.268
Duration of symptoms	≤3 m vs. >3 m	0.909	0.651–1.268	0.620	1.468	0.837–0.319	0.170	0.978	0.703–1.362	0.945
Radiotherapy	postoperative vs. definitive	0.592	0.845–0.415	0.004	0.517	0.846–0.270	0.006	0.601	0.854–0.423	0.004
Prolongation of radiotherapy	no vs. yes	1.509	1.024–2.223	0.036	0.477	0.466–1.223	0.010	0.634	0.934–0.431	0.020
Total dose	≤69 vs. >69 Gy	0.863	0.601–1.241	0.427	0.755	0.934–2.150	0.252	0.828	0.577–1.188	0.306
Chemotherapy	yes vs. no	0.721	1.006–0.517	0.053	1.417	0.985–0.314	0.099	0.696	0.966–0.501	0.029
Amifostin	yes vs. no	1.002	0.684–1.469	0.992	0.556	0.641–1.663	0.041	1.079	0.738–1.578	0.694
Weight loss	≤10% vs. >10%	1.004	0.697–1.446	0.497	1.032	0.659–1.526	0.072	0.940	0.653–1.351	0.534
Anemia	Hb ≥ 100 vs. Hb < 100 g/L	0.586	0.817–0.421	0.001	1.002	0.423–1.249	0.991	0.669	0.930–0.481	0.016
Hematotoxicity G3/4	yes vs. no	0.882	0.595–1.307	0.530	0.727	0.487–1.438	0.246	0.858	0.580–1.270	0.444
Feeding tube	yes vs. no	1.038	0.681–1.582	0.861	0.837	0.892–0.323	0.519	0.997	0.658–1.509	0.987
Pretreatment tracheostomy	yes vs. no	0.608	0.880–0.420	0.008	0.537	0.021–0.002	0.015	0.617	0.890–0.428	0.009
Response	CR vs. nonCR	0.119	0.188–0.076	<0.0001	0.007	0.784–1.803	<0.0001	0.026	0.052–0.013	<0.0001
Epoch	2002–2011 vs. 2012–2019	0.950	0.679–1.328	0.763	1.189	0.710–1.738	0.416	1.044	0.750–1.452	0.800

**Table 4 biomedicines-13-00417-t004:** Multivariate analyses for disease-free interval (DFI), overall survival (OS), and disease-free survival (DFS); only factors significant in univariate analysis were calculated. Abbreviations: ACE = adult comorbidity evaluation score; Hb = hemoglobin level; CR = complete response; HR = hazard ratio; CI = confidence interval.

Parameter	Groups	HR	95% CI	*p*-Value
**Disease free interval**				
Radiotherapy	postoperative vs. definitive	0.556	0.331–0.934	0.027
Prolongation of radiotherapy	no vs. yes	1.265	0.695–2.303	0.441
Amifostin	yes vs. no	1.299	0.723–2.333	0.381
Pretreatment tracheostomy	no vs. yes	1.252	0.708–2.217	0.440
Initial response	CR vs. non-CR	0.008	0.002–0.025	<0.0001
**Overall survival**				
Radiotherapy	postoperative vs. definitive	0.659	0.439–0.991	0.045
Prolongation of radiotherapy	no vs. yes	1.087	0.730–1.618	0.682
Pretreatment tracheostomy	no vs. yes	0.620	0.413–0.930	0.021
Anemia	Hb ≥ 100 vs. <100 g/L	0.637	0.449–0.906	0.012
Initial response	CR vs. non-CR	0.129	0.080–0.208	<0.0001
**Disease free survival**				
ACE	ACE 0–1 vs. 2–3	0.568	0.376–0.857	0.007
Radiotherapy	postoperative vs. definitive	0.649	0.430–0.883	0.039
Prolongation of radiotherapy	no vs. yes	1.167	0.766–1.777	0.472
Chemotherapy	yes vs. no	0.620	0.435–0.883	0.008
Pretreatment tracheostomy	no vs. yes	1.294	0.841–1.990	0.241
Anemia	Hb ≥ 100 vs. <100	1.281	0.881–1.863	0.194
Initial response	CR vs. non-CR	0.027	0.013–0.055	<0.0001

**Table 5 biomedicines-13-00417-t005:** Multinomial logistic regression model for overall survival (OS) and disease-free interval (DFI). Both nominal responses were discretized (0–1 year, 1–2 years, 2–3 years, 3–maximum years). This type of discretization is best for equal distribution of patients (median values for OS and DFI was around 2 years).

OS		*p*-Value	DFI		*p*-Value
Intercept coefficient (≤1 year/>1 year)	−3.049	<0.0001	Intercept coefficient (≤1 year/>1 year)	−2.728	<0.0001
Intercept coefficient (≤2 year/>2 year)	−1.431	<0.0001	Intercept coefficient (≤2 year/>2 year)	−2.118	<0.0001
Intercept coefficient (≤3 year/>3 year)	−0.878	0.008	Intercept coefficient (≤3 year/>3 year)	−1.961	<0.0001
Operation coefficient	0.968	0.006	Operation coefficient	1.296	<0.0001
Chemotherapy coefficient	0.980	0.003	Chemotherapy coefficient	0.736	0.016
Initial treatment response coefficient	3.301	<0.0001	Interval histology to radiotherapy	0.013	0.010

**Table 6 biomedicines-13-00417-t006:** Univariate logistic regression model for overall survival (OS) and disease-free interval (DFI) by Hypopharyngeal risk score. Both nominal responses were discretized (0–1 year, 1–2 years, 2–3 years, 3–maximum years).

OS		*p*-Value	DFI		*p*-Value
Intercept coefficient (≤1 year >1 year)	−4.133	<0.0001	Intercept coefficient (≤1 year/>1 year)	−4.074	<0.0001
Intercept coefficient (≤2 year/>2 year)	−2.484	<0.0001	Intercept coefficient (≤2 year/>2 year)	−3.050	<0.0001
Intercept coefficient (≤3 year/>3 year)	−1.902	<0.0001	Intercept coefficient (≤3 year/>3 year)	−2.860	<0.0001
Hypopharynx risk score	0.989	<0.0001	Hypopharynx risk score	1.174	<0.0001

## Data Availability

The data that support the findings of this study are not publicly available to ensure that the privacy of the research participants is not compromised. However, the data are available from the corresponding author upon reasonable request.

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
