# Peer review of "Prognostic Factors and Long-Term Outcome Prediction in Patients with Hypopharyngeal Carcinoma Treated with (Chemo)radiotherapy: Development of a Prognostic Model"

_biomedicines, 2025, doi:10.3390/biomedicines13020417_

Round 1

Reviewer 1 Report

Comments and Suggestions for Authors

The authors present a retrospective clinical study, regarding the prognostic factors and treatment of hypopharyngeal cancer. It is an interesting topic, that could add to the literature. However, there are some issues that need to be addressed, prior to consideration for publication.

  • The abstract is well structured and describes adequately the study.
  • The introduction provides a proper background.

There could be more reference to support what is mentioned in the first paragraph.

  • Concerning the methodology part, in my opinion the statistical data regarding the patients’ demographics, and treatment data as well as tables 1 and 2 should be mentioned in the results.
  • The results are adequately presented.

Please, correct “parametr” in Table 4.

  • In the discussion part the authors try to support their assumptions with notable references. Perhaps, it could be upgraded with some recent metanalyses and systematic reviews that are encountered in the literature, for example:

Panda S, Sakthivel P, Gurusamy KS, Sharma A, Thakar A. Treatment options for resectable hypopharyngeal squamous cell carcinoma: A systematic review and meta-analysis of randomized controlled trials. PLoS One. 2022 Nov 29;17(11):e0277460. doi: 10.1371/journal.pone.0277460. 

Tsai TY, Yap WK, Wang TH, Lu YA, See A, Hu YF, Huang Y, Kao HK, Chang KP. Upfront Surgery Versus Upfront Concurrent Chemoradiotherapy as Primary Modality in Hypopharyngeal Squamous Cell Carcinoma: A Systematic Review and Meta-Analysis. J Otolaryngol Head Neck Surg. 2024 Jan-Dec;53:19160216241293633. doi: 10.1177/19160216241293633.

 The quality of English language is quite good. A review by a native English speaking editor/service check for a number of minor but significant grammatical/syntax issues would be appreciated.

We look forward to your revisions.

Reviewer 2 Report

Comments and Suggestions for Authors

The purpose of this study was to evaluate outcomes and prognostic factors for patients with head and neck cancer treated with radiation therapy. The authors conducted an analysis of patients treated between 2002 and 2020 and examined prognostic factors. Several issues need to be addressed before publication.

Major comments:

  1. The long study period (2002-2020) may lead to changes in treatment methods and technologies. This should be addressed in the discussion.
  1. The long study period (2002-2020) may introduce significant time bias due to changes in treatment protocols (e.g., introduction of new drugs such as cetuximab), advances in diagnostic techniques, and improvements in radiotherapy techniques. Authors should report major changes in treatment protocols during the study period. Also, consider stratified analysis by early and late study periods or multivariate analysis by including year of diagnosis as a covariate.
  1. Although internal validation of the prognostic model was performed, external validation is lacking. This should be discussed in detail as it is crucial to assess the generalizability of the model.
  1. The use of KM methods may be problematic due to competing risks, especially for patients who died before receiving radiotherapy. Authors should consider using competing risk analysis or at least discuss this limitation and its potential impact on the results. It is important to report the number and characteristics of patients who died or were unable to receive radiotherapy before the start of treatment, so that readers can assess the potential impact of competing risks.
  1. The authors used univariate logistic regression to validate their scoring system. They should explain why they chose this method over multivariate analysis and discuss potential limitations of this approach.

Minor comments:

  1. Please add the number of patients at risk in Figures 1-4.
  1. Please add explanations for abbreviations (e.g., ACE) in footnotes for Tables 3 and 4, and spell out all abbreviations in each table.

Round 2

Reviewer 2 Report

Comments and Suggestions for Authors

Thank you very much for this opportunity of re-review the manuscript. I confirm all concerns have been addressed by the authors.